# Microstructural Modification Hardness and Surface Roughness of Hypereutectic Al-Si Alloys by a Combination of Bismuth and Phosphorus

Porawit Jiandon ⓘ and Sukangkana Talangkun *ⓘ

Faculty of Engineering, Khon Kaen University, Khon Kaen 40002, Thailand; porawit_j@kkumail.com
* Correspondence: sukangkana@kku.ac.th; Tel.: +66-43-362145-6

**Abstract:** This research aimed to investigate the effects of bismuth (Bi) and phosphorus (P) addition on the microstructure and roughness of a B390 hypereutectic Al-Si alloy. Microstructural variation of as-cast alloys showed that P is an effective refiner of primary Si and its distribution, while Bi has the opposite effect. The liquidus temperature increased with greater amounts of P, resulting in a wider solidification range. The solidus temperature increased with Bi content, resulting in a shortened solidification range. The benefit of Bi on eutectic silicon was through changing its morphology from fibrous to lamellar structures. It was observed that both primary Si and eutectic Si were modified when both P and Bi were added to the alloys. For effective refinement, the size of primary Si can be reduced by 50.6% and, the P content should not be less than 0.1 wt% with a Bi content of 0.5–1.0 wt%. Furthermore, an intermetallic phase size reduction also occurs. P addition results in a significant (5.85%) increase in hardness. Adding Bi (up to 1.0 wt%) together with P (up to 0.1 wt%) did not reduce the effect of P on the hardness of the modified alloys. The most effective reduction in roughness achieved during cutting at both low and high speeds was 26.5% and 57.7%, respectively, from modifying the B390 alloy using 0.1 wt%P with 1.0 wt%Bi.

**Keywords:** phosphorus; bismuth; hypereutectic Al-Si alloys; primary silicon; modification



## 1. Introduction

Hypereutectic Al-Si alloys are widely used in the automotive and aerospace industries due to their excellent castability, low coefficient of thermal expansion, and high strength-to-weight ratio [1]. Aluminium alloy B390 is one of the hypereutectic Al-Si alloys used for both die and permanent mold castings. This alloy is typically employed for engine cylinder blocks and compressors. The silicon particles contained in this alloy provide good wear resistance and thermal stability. In a conventional casting, however, the formation of coarse primary Si with irregular and polygonal shapes, and coarse flakes of eutectic Si were observed, negatively affecting the mechanical properties and machinability of the hypereutectic Al-Si alloys [2]. Over the past several decades, most researchers have attempted to develop hypereutectic Al-Si alloys using modifiers such as sodium (Na), phosphorus, strontium (Sr), bismuth antimony (Pb), and intermetallic compounds. This partway alters the size, shape, and distribution of the primary silicon phase and the eutectic Si phase, meeting the requirements of industrial applications. Surface roughness (Ra), which is a significant indicator of workpiece quality, has been generally utilized for evaluating the machinability of alloys. This is especially relevant for applications demanding excellent surface quality in Al-Si casting alloys for the automotive industry and other applications.

Modification processes are generally regarded as effective techniques to improve the mechanical properties of hypereutectic Al-Si alloys. Helal et al. (2012) suggested that the primary Si particles in microstructure should become a uniform distribution of fine particles [3]. Phosphorus is one of the most effective modifiers for primary silicon

refinement [4]. Generally, phosphorus is introduced into the melt as master alloys such as Al–Cu/Fe–P [5], Al–Si–P [6], Al–Zr–P [7], Al–P [8], and Cu–P master alloys [9]. The refinement mechanism relates to the nucleation of the fine aluminium phosphide (AlP) compounds after adding phosphorus master alloys into the melt. During the solidification process, these AlP nuclei act as sites of nucleation for the primary Si phase. Zhang et al. (2008) found that the modification mechanism of Cu-P on primary Si in the castings resulted from heterogeneous nucleation around AlP particles [10]. Wu et al. (2009) showed that AlP particles were formed after adding the Si–P master alloy into Al melt. Some of them can act as the nuclei of primary Si phases during the solidifying process, and thus refine the primary Si phase [11]. The evidence found in the work of Hongshang et al. (2009) showed the presence of AlP compounds in the centre of primary Si due to adding a phosphorus master alloy to the melt [12]. The formation of AlP leads to an increase in the nucleation of primary Si and a concurrent reduction in the size of the primary Si phase [13]. AlP can form quickly after adding a Cu-P alloy into the melt because the Cu-P master alloy is easy to melt at a high pouring temperature (800 °C) [10]. Although Cu-P alloy is considered an economical modifier [10], it showed unstable modification and was not environmentally friendly [11].

It is important to note that the excellent mechanical properties and machinability of hypereutectic Al-Si alloys depend on the morphology and distribution of both primary Si and eutectic Si [14]. Therefore, both of them need to be effectively modified and refined simultaneously. Kamiya et al. (2008) studied the effect of eutectic Si and primary Si on the machinability of Al-Si alloy castings and their investigation indicated that fractured primary Si was responsible for the increased surface roughness and that tool wear increased with the amount of eutectic Si [15]. It has been recognised that strontium is an effective modifier of morphology and the distribution of the eutectic phase in Al-Si alloys [16]. However, an increase in the Sr content is likely to enhance the aluminum alloy adhesion to the face of cutting tools, resulting in increased built-up edge formation and new surface roughness, as well as increased cutting and feed forces. These consequences are not favourable in the machining process. Barzani et al. (2013) have shown that, among modified alloys, a strontium-containing workpiece (base ADC12 alloy) exhibited the highest surface roughness, highest cutting force, and highest feed force compared to the base alloy, bismuth-containing and antimony-containing alloys [17]. As an alternative, Bi is currently used as an effective modifying agent for eutectic Si in Al-Si alloys. The addition of Bi in the casting process is suitable for components that require machining. Bi offers several advantages for improved machinability of Al-Si alloys, including promoting chip breaking and lubrication by forming a soft low melting point phase that decreases friction between the chips and cutting tool as well as reducing the need for cutting fluids. The presence of Bi is also known to increase the machining speed [18]. Barzani et al. (2013) proposed that the bismuth-containing alloy (base ADC12 Alloy) seems to produce the best surface finish during turning [17]. Moreover, it contributes to the prevention of hot cracking in Al-Mg alloys [19]. Bi is recognised as a modifier of eutectic phase morphology as shown in the work of Farahany et al. (2013), where the transformation of a flake to the lamellar structure was observed after the addition of Bi in ADC12 alloy [20]. Samuel et al. (2016) observed large acicular Si particles predominating in the microstructure after Bi addition in an Al-Si B319 alloy [18].

Recently, there have been attempts to modify the microstructure of hypereutectic Al-Si alloys with various elements. Chong et al. (2007) studied the modification of a hypereutectic Al-20Si alloy by P and RE complex addition [21]. It was found that the addition of 0.08%P and 0.60% rare earth (RE) caused the optimal microstructure, a refined primary Si and fine fibrous eutectic Si. In addition, Zuo et al. (2013) studied the modification of hypereutectic Al–30Si alloys using P and Sr [4]. They pointed out that primary Si nucleated on the substrate of AlP and thus its significant refinement with average size decreased from 203.8 μm to approximately 32.8 μm. Sr also modified eutectic Si to a fine fibrous microstructure. Lin et al. (2019) examined modifying Al–18Si–4Cu–0.5Mg alloy by

the addition of La–Ce rare earth elements [22]. The results indicated that primary Si was refined from coarse block-like and irregular polygonal shapes into fine flaky shapes, while eutectic Si particles were altered from coarse and needle-like forms into fine and rod- or coral-like shapes.

Although Bi has been used by many researchers as an effective modifying agent for eutectic Si which enhances machinability in Al-Si alloys, their study focused only on hypereutectic and near-eutectic Al-Si alloys groups [1,17,18,20]. It has been shown that the refinement of coarse fibrous and large needle eutectic Si to fine fibrous or lamellar phases can be achieved using either Bi or Sr. Nevertheless, the Sr modifier may not be suitable for parts that require machining since it seems to reduce the machinability of Al-Si alloys. Additionally, P is recognised as an excellent modifier for improvement in the shape and distribution of the primary Si phase as it promotes the fine and uniform distribution of primary Si, leading to good mechanical properties. However, eutectic modifiers such as Bi do not interact with primary Si in the same way as they do with the eutectic phase. This is because the formation of these phases occurs at different temperatures.

Modification of primary Si together with eutectic Si in the casting of hypereutectic Al-Si alloys with a suitable combination of P and Bi agents has not yet been well defined in the literature. Furthermore, the relationship between the modified microstructure and surface roughness after cutting is not yet clear. Therefore, the aim of this research is to investigate the effects of both P as a primary Si modifier coupled with Bi as eutectic Si modifiers on microstructure, hardness, and surface roughness of the hypereutectic aluminium alloy B390 (JIS ADC14).

## 2. Materials and Methods

### 2.1. Materials

Aluminium alloy B390 ingots and rods of Cu-P alloys were supplied by DAIKI Aluminium Industry (Thailand) Co., Ltd. (Chon Buri, Thailand) and Harris Products Group (Mason, OH, USA), respectively. Pure bismuth (99.99%) with a needle shape was supplied by Vital Materials, Co., Ltd. (Guangzhou, China). The chemical composition of the B390 alloy was examined using an emission spectrometer (ARL 3460 OE Spectrometer, Thermo Scientific, Waltham, MA, USA). The results are shown in Table 1. The chemical composition of the B390 alloy conformed to Aluminium Association Standard (AA) [23].

**Table 1.** Chemical composition of the specimen used in this study (wt%).

| | Alloying Elements (wt%) | | | | | | | | |
|---|---|---|---|---|---|---|---|---|---|
| **Composition** | Si | Fe | Cu | Mn | Mg | Zn | Ni | Ti | Al |
| Standard | 16–18 | ≤1.3 | 4–5 | ≤0.5 | 0.45–0.65 | ≤1.5 | ≤1.5 | ≤1.5 | Bal. |
| Received B390 | 18.3 | 0.9 | 4.8 | 0.3 | 0.5 | 0.5 | 0.06 | 0.027 | Bal. |

### 2.2. Casting

The chemical composition of the alloys fabricated in this experiment was categorized into eight groups as listed in Table 2. In this process, the B390 alloy was initially melted in an alumina crucible in a furnace. After the process temperature reached 600 °C, the required quantity of Bi was introduced to the molten alloy. The temperature of molten liquid was subsequently increased to 710 °C, and the mixture was quickly agitated to ensure full melting of added Bi at that temperature. Then, Cu–7%P alloys were added to the melt, mechanically stirred, and held for 15 min. The molten alloy was then poured at 710 °C into a metal mold preheated to approximately 200 °C. The metal mold used in this experiment is shown in Figure 1.

**Table 2.** Chemical compositions of the modified alloys (wt%).

| Alloy | Modifiers | | Bi:P Ratio |
|---|---|---|---|
| | Bi(wt%) | P(wt%) | |
| B390-0.5Bi | 0.5 | 0 | 0.5:0 |
| B390-1.0Bi | 1.0 | 0 | 1.0:0 |
| B390-0.05P | 0 | 0.05 | 0:0.05 |
| B390-0.1P | 0 | 0.1 | 0:0.1 |
| B390-0.05P-0.5Bi | 0.5 | 0.05 | 0.5:0.05 (10:1) |
| B390-0.1P-0.5Bi | 0.5 | 0.1 | 0.5:0.1 (5:1) |
| B390-0.05P-1.0Bi | 1.0 | 0.05 | 1:0.05 (20:1) |
| B390-0.1P-1.0Bi | 1.0 | 0.1 | 1:0.1 (10:1) |

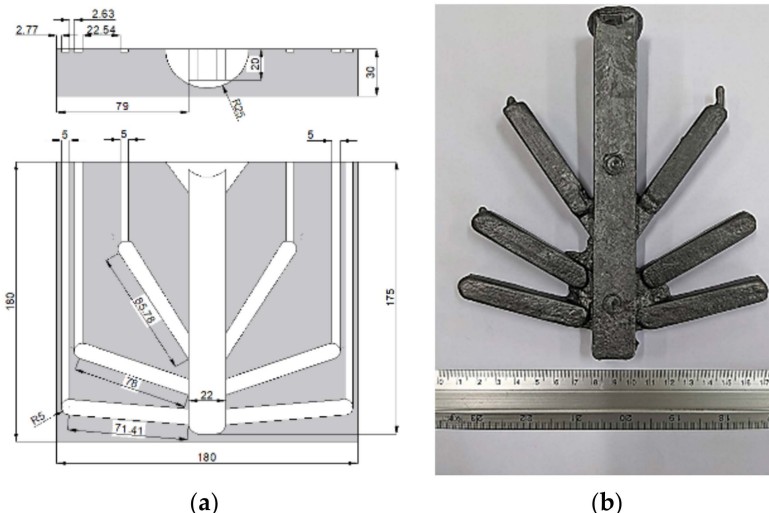

(**a**)          (**b**)

**Figure 1.** (**a**) Steel mould (scale: mm) and (**b**) as-cast sample.

### 2.3. Metallography

Specimens for metallographic microstructure observation were in the as-cast condition. The microstructural analyses of these samples were performed using an optical microscope (Olympus BX60, Olympus Optical Co., Ltd., Tokyo, Japan) and scanning electron microscope (LEO 1450, Jena, Germany) and field-emission scanning electron microscope (TESCAN model MIRA, Tescan Orsay Holding, a.s., Brno—Kohoutovice, Czech Republic). An energy dispersive spectrometry (EDS) investigation was also carried out. The size measurements were made using the horizontal Feret's Diameter method [24]. The results are presented as an average of 100 measurements. The distribution of the primary Si phase was measured using the point counting method [25].

### 2.4. Differential Scanning Calorimetry

The melting behavior of all samples was observed using a DSC (Hitachi model DSC7020, Hitachi High-Tech Science Corporation, Tokyo, Japan). A 10 mg was heated in a ceramic pan with a heating rate of 5 °C/min under a nitrogen flow of 30 mL/min. The solidus and liquidus temperatures were analysed.

### 2.5. Hardness and Surface Roughness Tests

The casts were sectioned, mechanically ground, and polished. Rockwell hardness tests (HRB) [26] were performed (ERGOTEST DIGI 25 RS, GALILEO Durometria, Antegnate, Italy). The results are presented as an average of three measurements. For surface roughness (Ra), first, each workpiece was subjected to face milling (CNC Mikron VCE 750, Mikron Group, Mühlebrücke, Switzerland) using an end mill 12EH402120 (D12) type 4FL, YG-1 Co., Ltd., Incheon, Korea). Machining parameters recommended by the cutting tool

manufacturer are shown in Table 3. After that, the machined surface was subjected to a surface roughness test (Ra) using the SURFTEST SJ 310 (Mitutoyo Corporation, Tokyo, Japan). The testing parameters were a Gaussian filter setting, a cut-off of 0.80 mm, and a measurement length of 4.00 mm.

**Table 3.** Face milling parameters.

| No. | Face Milling Parameters | | | |
| --- | --- | --- | --- | --- |
| | Spindle Speed (rpm) | Cutting Speed (m/min) | Depth of Cut (mm) | Feed Rate (mm/min) |
| 1 | 2080 | 78 | 0.2 | 1020 |
| 2 | 2600 | 98 | 0.2 | 1020 |

## 3. Results and Discussion

### 3.1. B390 Initial Microstructure

The microstructure in the cross-sectional area of the cast B390 is shown in Figure 2 and it was dendritic in nature. The B390 alloy consisted of dendritic α-Al, primary Si, and eutectic phases. The initially unmodified primary Si phase, which was formed above the liquidus temperature at approximately 670 °C [27], exhibited a large polygonal morphology with an average size of 78.90 μm, as indicated by the arrows in Figure 2a. This phase was agglomerated and inhomogeneously dispersed in the microstructure. Additionally, a coarse fibrous phase was present in the area of the eutectic phase, as shown in the optical and SEM images of Figures 2b and 3, respectively. This phase included $Al_{15}(Mn, Fe)_3Si_2$ (like Chinese script) in a pre-eutectic region. The co-eutectic phase including an Si phase (grey color) $Al_5FeSi$ (needles), $Al_5Mg_8Cu_2Si$ (bulk), a small amount of $Mg_2Si$ (black) and $Al_2Cu$ (fine particles) [27,28].

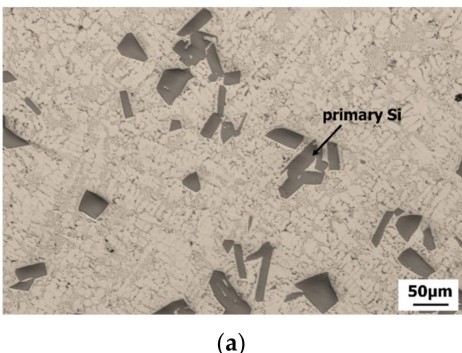
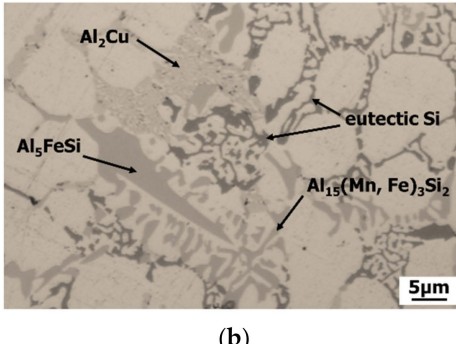

(**a**)　　　　　　　　　　　　　　　　　(**b**)

**Figure 2.** Microstructure of the initial B390 showing: (**a**) a dendritic α-Al phase, primary Si, and (**b**) eutectic phases.

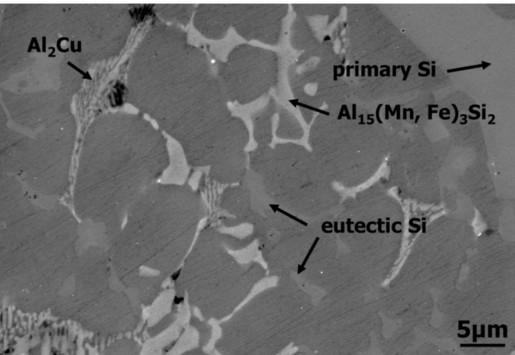

**Figure 3.** SEM image of the initial B390 showing intermetallic phases.

### 3.2. Effect of Bi and P on Microstructure

#### 3.2.1. Primary Si Refinement

Figure 4 shows the average diameter of primary Si measured using Feret's method along with an average phase distribution. Approximately 20% of the primary Si phase, with an average size of 78.9 μm, was observed in the unmodified alloy. By adding 0.5 wt%Bi, the average size of the primary Si phase slightly decreased to approximately 70.28 μm with a similar density distribution as an unmodified B390. However, when increasing the Bi concentration to 1.0 wt%, the average size of the primary Si phase increased to 81.16 μm and was less well distributed. Alternatively, it was observed that by increasing the P content to levels between 0.5–0.1 wt%, the average size of the primary silicon phase in the B390 alloy gradually decreased from 24.36 μm to 18.5 μm, and the distribution of this phase throughout the cross-section of the sample was improved. It has been suggested that the P content has a greater role than the Bi content in the improvement of size reduction and the distribution of primary Si. The most effective refinement of primary Si was obtained from a 0.1 wt%P modified alloy, which had an average size of 18.15 μm, equivalent to a 76% reduction in size. Moreover, simultaneous increases of the Bi and P contents at various proportions in the B390 alloy led to further decreases in the average size of the primary Si phase, compared to unmodified B390 and the B390 alloy modified with Bi. However, the size reduction and phase distribution of both P with Bi were less than that of the P modification. The sizes of the primary silicon phase of B390-0.05P-0.5Bi, B390-0.1P-0.5Bi, B390-0.05P-1.0Bi and B390-0.1P-1.0Bi were 60.07, 38.96, 62.48 and 49.29 μm, respectively. It is notable that with an increase in the Bi:P ratio, the size of the primary silicon phase increased. Even though the ratios of Bi:P of both B390-P0.05-0.5Bi and B390-0.1P-1.0Bi were the same, 10:1, the average size of B390-P0.05-0.5Bi was greater than that of B390-0.1P-1.0Bi.

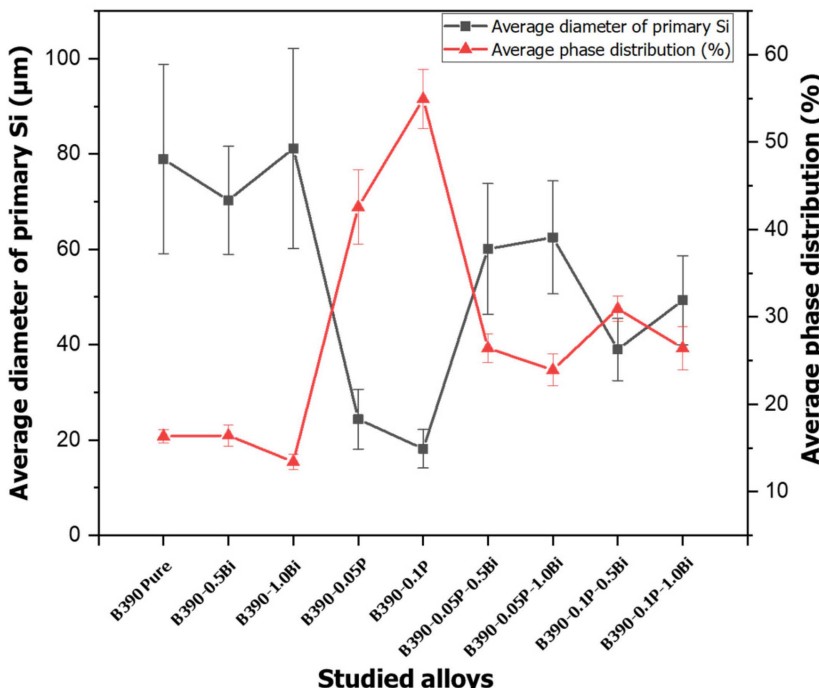

**Figure 4.** Average diameter and phase distribution of the primary Si of B390 and modified B390 alloys.

#### 3.2.2. Primary Si Morphology

Microstructure of modified B390 with 0.5 wt%Bi and 1.0 wt%Bi addition are shown in Figures 5a and 6b, respectively. It should be noted that the distribution of primary Si appears more clustered compared to the initial B390. The influence of bismuth on the cooling curves appears as a depression of the nucleation temperature, $T_N$, which is consistent with the experimental result of Farahany et al. (2014), allowing more time for

pre-eutectic primary Si, which had been formed earlier, to grow and agglomerate until $T_N$ is reached [29]. Additionally, Bi cannot form compounds with aluminium that act as nucleation sites for primary silicon. Alternatively, compared to the initial B390 and the Bi modified sample, the size of primary Si decreases with increased levels of P, as shown in Figure 5c,d. Furthermore, with increasing P content, the morphology of primary Si was drastically changed from clustered large bulky polygons to scattered fine ones. It should also be noted that the microstructure of the B390-P samples appears more uniformly distributed, with a higher density of primary Si. This indicates a more even nucleation of phosphide particles in the melt, i.e., AlP. Furthermore, AlP particles and primary Si have the same FCC crystal structure. When AlP randomly nucleates during melting, it provides nuclei for the primary Si upon cooling [9], increasing the nucleation rate during solidification. Additionally, phosphorus appears to have some influence on the cooling curves by increasing undercooling and $T_N$. This is because the AlP in the melt may nucleate in regions with a higher P concentration and combine with Si particles, resulting from the higher reaction temperature. As a result, the size of the small primary Si decreases.

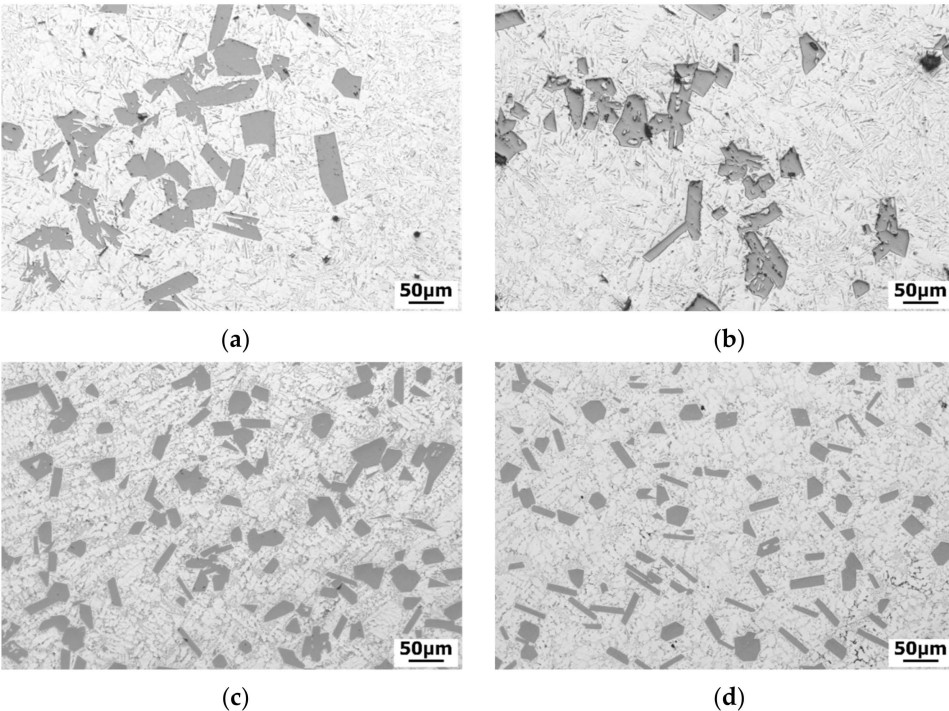

**Figure 5.** Microstructures of (**a**) B390-0.5Bi (**b**) B390-1.0Bi (**c**) B390-0.05P and (**d**) B390-0.1P.

The morphology of primary Si changes with the ratio of P to Bi. Modifications were examined using optical micrographs, as shown in Figure 6. In a comparison of unmodified B390 and Bi modified specimens, when both P and Bi were added, the size of primary Si decreased and Si appeared to be distributed more evenly. The series of modification experiments indicate that Bi and P have different effects on the formation of primary Si. From Figures 5a and 6a, at the same level of Bi, 0.5 wt%, the addition of 0.05 wt%P can reduce the size of the primary Si phase, but still, show agglomerates and clusters of Si (Figure 6a). Figure 6b shows that when the phosphorus content is increased to 0.1 wt%, the size of the primary Si is further reduced and there is a more even distribution of this material compared to specimens B390-0.5Bi (Figure 5a) and B390-0.05P-0.5Bi (Figure 6a). When increasing the amount of Bi to 1.0 wt% and P to 0.05 wt%, it was found that when the ratio of Bi:P is higher, i.e., 20:1, the size of primary silicon is larger, as shown in Figure 6c. In Figure 6d, although the size of the primary Si decreased when 0.1 wt%P was added, it was still larger and less well distributed compared to the sample modified with only P (Figure 5d).

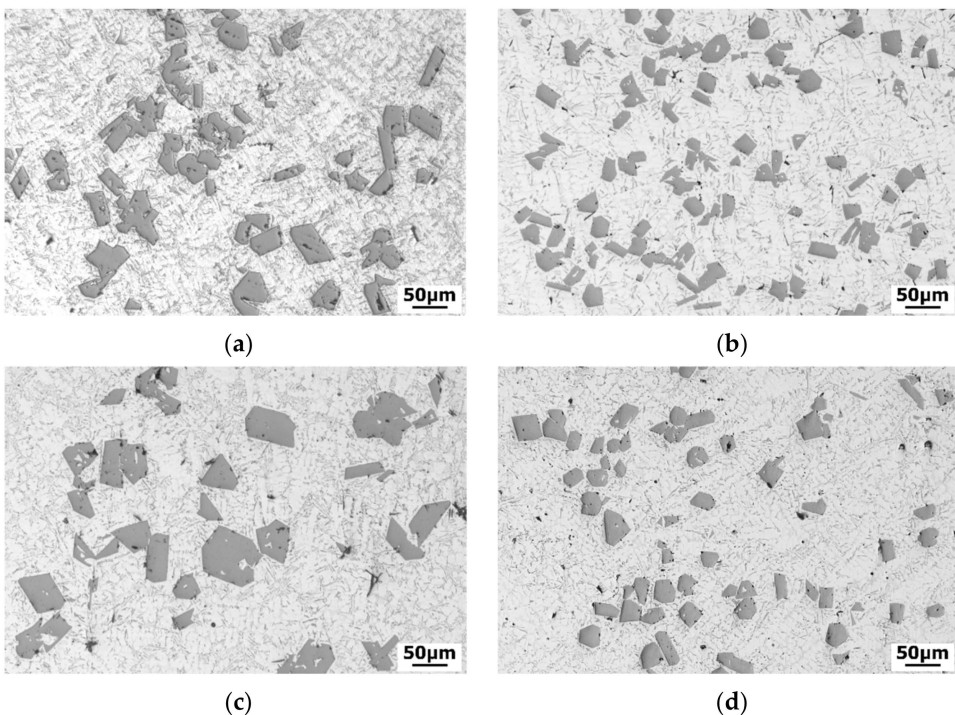

**Figure 6.** Optical micrographs showing the effect of Bi and P on primary silicon of (**a**) B390-0.05P-0.5Bi (**b**) B390-0.1P-0.5Bi (**c**) B390-0.05P-1.0Bi and (**d**) B390-0.1P-1.0Bi.

### 3.2.3. Eutectic Silicon and Intermetallic Compounds

When modified with 0.5 wt%Bi, the eutectic Si appeared lamellar, as shown in Figure 7a, however, the size of the eutectic phase was reduced with increasing Bi content. When the Bi content reached 1.0 wt% in the melt, a refinement of Al-Si flakes was observed, Figure 7b, resulting in the formation of lamellar structures. There is also evidence that the inter-flake spacing decreased with the Bi content, an indication of refinement. The modification mechanism is that the Al-Si eutectic is accompanied by a depression in the growth temperature, $T_G$ [30], and the low melting point is just above 271 °C for Bi. As $T_G$ decreases, leading to a nucleation rate increase. This results in a reduction of size and morphology change of the eutectic phase [31]. Additionally, an increase in the amount of Bi reduces $T_G$, yielding a greater level of refinement. This effect is similar to that when Sr is used as a modifying agent. It is established that the addition of certain electropositive elements, such as Sr and Na, will completely change the morphology of the material by adsorbing it and thereby disturbing the growth mechanism of Si through the Twin Plane Re-entrant Edge (TPRE) and Impurity Induced Twinning (IIT) mechanisms. This results in a change of the eutectic silicon crystals from large flakes into a fibrous structure resembling some types of seaweed and coral. An optimal ratio of the ad-atoms to the silicon atoms of around 1.65 induces a high degree of twinning. In the case of Sr, the nucleation and growth temperatures are around 10 °C lower [27]. Despite Bi being a post-transition metal, it can depress the TG of the aluminium alloy 356 by approximately 3 °C [31]. The empirical atomic radii of Bi, Sr, Na, and Si are 156, 215, 186, and 111 pm, respectively, giving the atomic ratios of Bi:Si, Sr:Si, and Na:Si as 1.41, 1.93, and 1.67, showing that Bi fulfils the atomic ratio criterion reasonably well.

From Figure 7c,d, it can be seen that P addition has no effect on the shape and distribution of the eutectic Si. This result agrees with the work of previous researchers (Zuo et al. (2009)) since AlP is not acting as the nucleus of eutectic Si [6]. It is established that β-(Al, Si, Fe) is a major phase that provides nucleation sites for eutectic Si [32].

However, the morphology of eutectic Si changed when P and Bi were added together, as shown in Figure 8. When P of 0.05 and 0.1 wt% was added to B390-0.5Bi, eutectic Si became lamellar as shown in Figure 8a,b, respectively. Compared to the unmodified B390,

the eutectic Si of the P-modified specimens changes from a coarse fibrous to a lamellar shape and is more evenly distributed than in the 0.5 wt%Bi modified alloys. Furthermore, when Bi content reached 1.0 wt%, eutectic Si appears in a fine lamellar shape and has a more uniform distribution, as shown in Figure 8c,d, respectively.

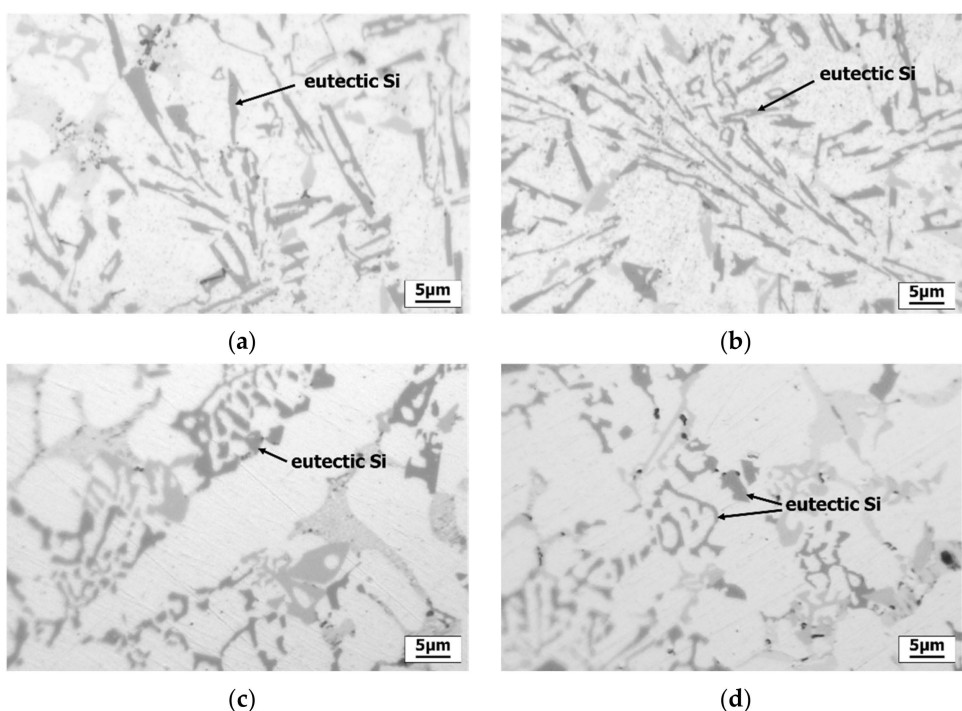

**Figure 7.** Optical micrographs showing the effect of Bi and P on the eutectic phases of (**a**) B390-0.5Bi (**b**) B390-1Bi (**c**) B390-0.05P and (**d**) B390-0.1P.

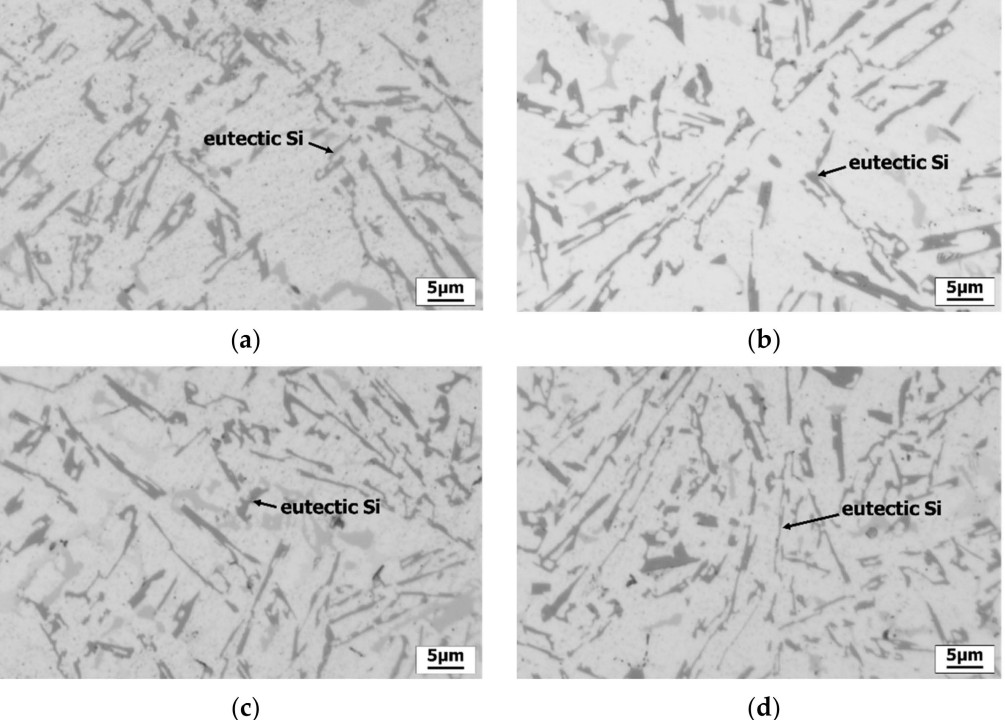

**Figure 8.** Optical micrographs showing the effect of Bi and P on eutectic phases of (**a**) B390-0.05P-0.5Bi (**b**) B390-0.1P-0.5Bi (**c**) B390-0.05-1.0Bi, and (**d**) B390-0.1P-1.0Bi.

Further explanation of the effects of the combination of Bi and P on microstructure can be demonstrated using EDS mapping, as shown in Figure 9. In Figure 9a, unmodified B390 showed that eutectic Si (blue dendritic) likely forms around the bulky primary Si. In Figure 9b, the matrix of the B390-0.1P sample around the small primary Si retains a high concentration of Si. During cooling, AlP particles (as shown as red dots) act as nucleation sites for primary Si. This induces an even distribution of primary Si, followed by the formation of dendritic structures that cause a high concentration of Si atoms around the primary Si and in the inter-dendritic area. In Figure 9c, it can be seen that the Si concentration is depleted around the primary Si. Since the Bi content facilitates the distribution of Si throughout the matrix, eutectic Si can randomly form lowering the concentration of Si in the matrix. In a Bi-P modified sample, both primary Si and eutectic Si were evenly dispersed throughout the matrix evidenced by a homogenous distribution of Si, as shown in Figure 9d.

In the case of intermetallic compounds, as seen in Figure 10 (bright areas), it was found that the unmodified specimen consisted of a large number of interconnections in the intermetallic compounds have mainly coarse dendritic network, as shown in Figure 10a. With additional Bi, the intermetallic phase was fine dendritic and more dispersed compared to the unmodified and P conditions, as shown in Figure 10b. Generally, adding bismuth will result in a reduction in $T_G$, affecting the Al-Si eutectic and intermetallic compound growth temperature since the temperature range of intermetallic compound growth is in the same range as the Al-Si eutectic [27]. In P addition, it can be seen that the intermetallic phases are mainly dendritic and were non-uniformly distributed in the pre-eutectic area due to P continuig to influence the distribution of silicon concentrations throughout the melt, resulting in the area of intermetallic phases formation being small and obstructed by primary Si small size and the distribution. In particular, the intermetallic phases that contain Si such as $Al_{15}(Mn, Fe)_3Si_2$, $Al_5FeSi$, and $Al_5Mg_8Cu_2Si_6$ were finer and more evenly distributed throughout the alloy, as shown in Figure 10c. In Figure 10d when both P and Bi were added, the size of the intermetallic phase was non dendritic, fine and better dispersed, different from the unmodified, Bi and P condition due to influence by both P and Bi modification, as shown in Figure 10c.

It is confirmed by microstructural analysis (SEM-EDS), and according to the binary phase diagrams Mg–Bi system [33,34], the presence of Bi in the microstructure with two forms, consisting of the Bi particles and intermetallic $Mg_3Bi_2$ phase, resulted from the Bi addition. The chemical reaction between Bi and Mg leads to the formation of the intermetallic $Mg_3Bi_2$ phase with angular or hexagonal particles [34]. With the Bi particles presence on the microstructures of Bi and both P and Bi addition, Bi particles present as small round particles found inside and at the surface of primary Si (arrowed in Figure 11c). The particles with Bi content can be found in the area without Mg content, as shown in Figure 11. This result is in agreement with the work of Biswas et al. (2019) [35]. While the intermetallic $Mg_3Bi_2$ phases were found along the grain boundaries due to the area with Mg-rich content, this result is in agreement with Rečnik et al. (2021) [33]. Similarly, in this experiment, $Mg_3Bi_2$ (hexagonal anti-$La_2O_3$ type structure with the space group of P3m1 [36]), was found adjacent to $Al_2Cu$ (tetragonal structure), as shown in Figure 12.

### 3.3. Effect of Bi and P on Solidification Range

Differential scanning calorimetry (DSC) curves of the B390 alloys are shown in Figure 13. Table 4 lists the measured solidus and liquidus temperatures of each alloy. The unmodified B390 alloy had a solidification range from 507 to 647 °C. It was found that when phosphorus was added at 0.05 wt% and 0.1 wt% to the B390 alloy, the liquidus temperature increased by 5.1 and 18.2 °C, respectively, compared to the unmodified alloy. The solidus temperature remained unchanged, i.e., the solidification range increased. This indicates the presence of primary Si at a higher temperature. Alternatively, when bismuth at 0.5 wt% and 1.0 wt% were added, it was found that the liquidus temperature was similar to the unmodified alloy, while the solidus temperature increased by 8.7 and

10.7 °C, respectively. An increase in the solidus temperature resulted in a modified eutectic Si and intermetallic compounds due to a reduction of the eutectic temperature range. In the case where both P and Bi were added, samples exhibited changes in both the solidus and the liquidus temperatures. These indicate modifications of both primary Si and eutectic Si from an increased solidus temperature by Bi and liquidus temperature by P on the solidification reaction.

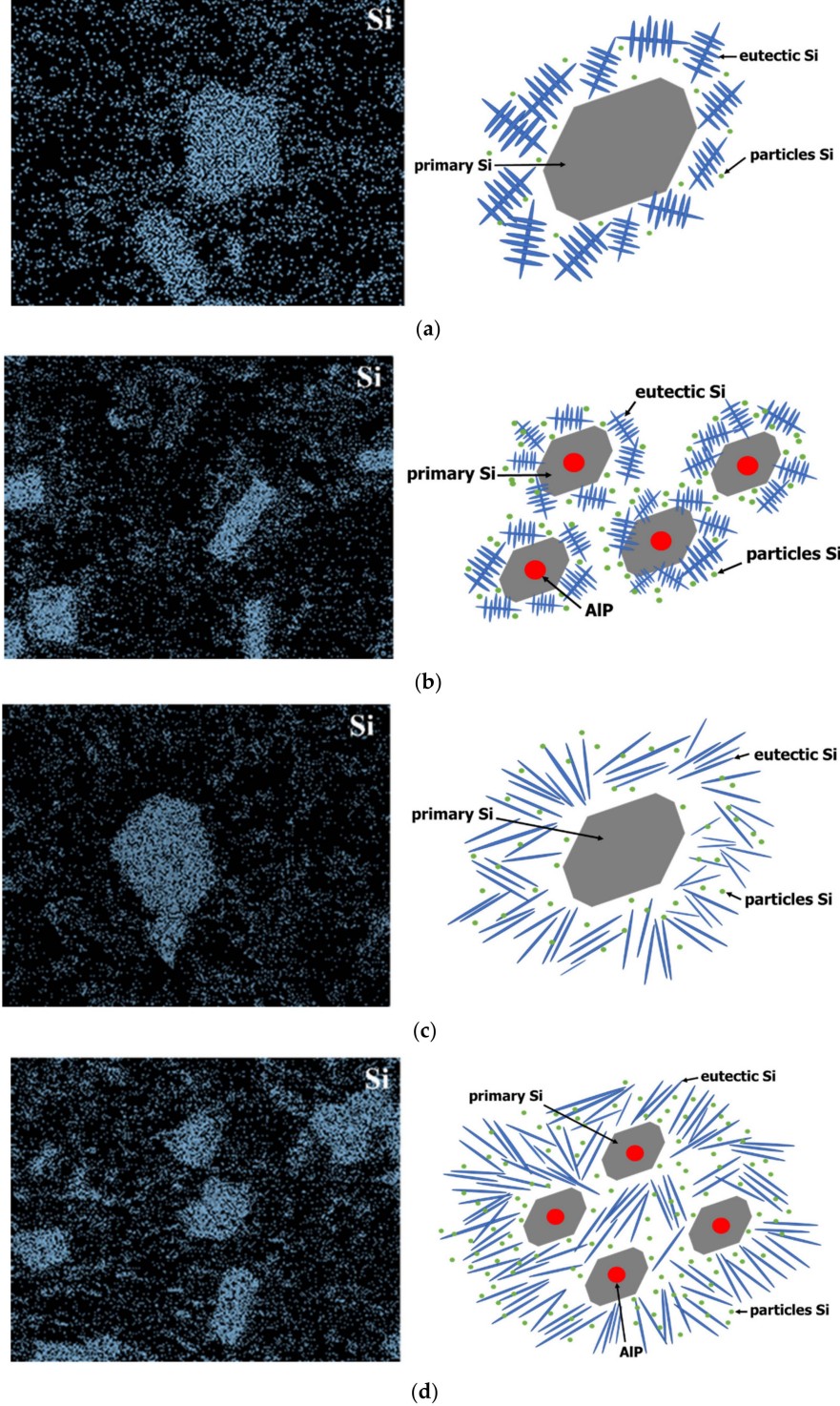

**Figure 9.** EDS mapping shows the concentration and dispersion of silicon along with the intermetallic phase model in (**a**) unmodified B390 (**b**) B390-0.1P (**c**) B390-1.0Bi, and (**d**) B390-0.1P-1.0Bi.

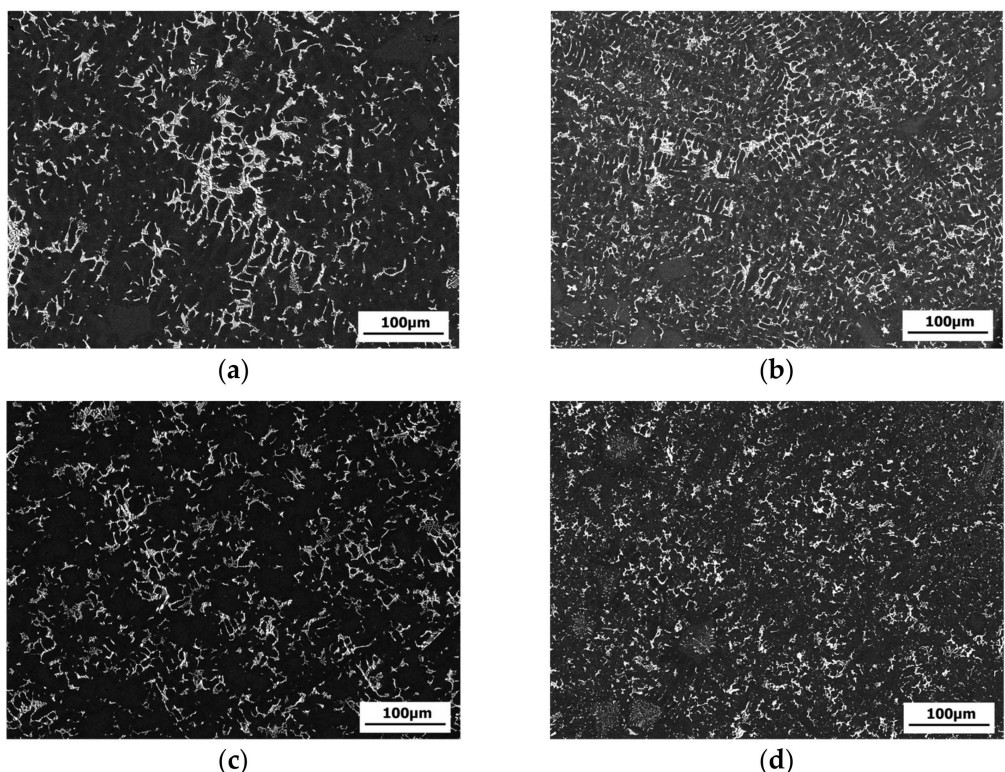

**Figure 10.** SEM images show the morphology of intermetallic phases (**a**) unmodified B390 (**b**) B390-1.0Bi and (**c**) B390-0.1P (**d**) B390-0.1P-1.0B.

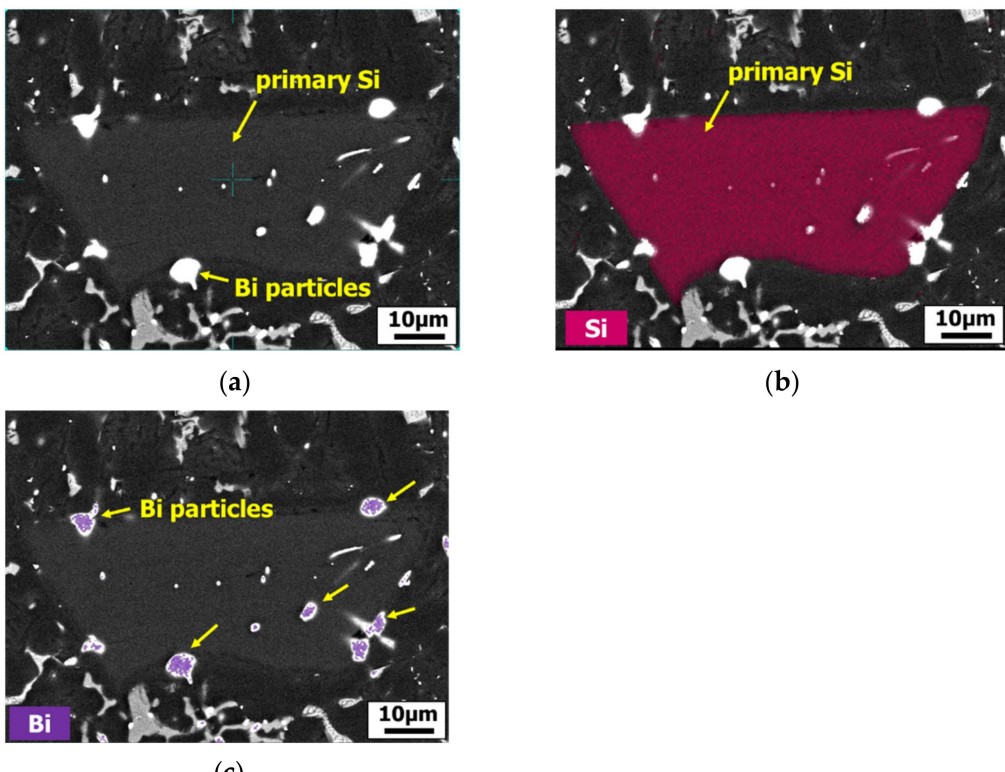

**Figure 11.** Elemental mapping shows on B390-1Bi (**a**) Bi particles are inside and at the surface of primary Si (**b**) elemental mapping of Si, and (**c**) elemental mapping of Bi.

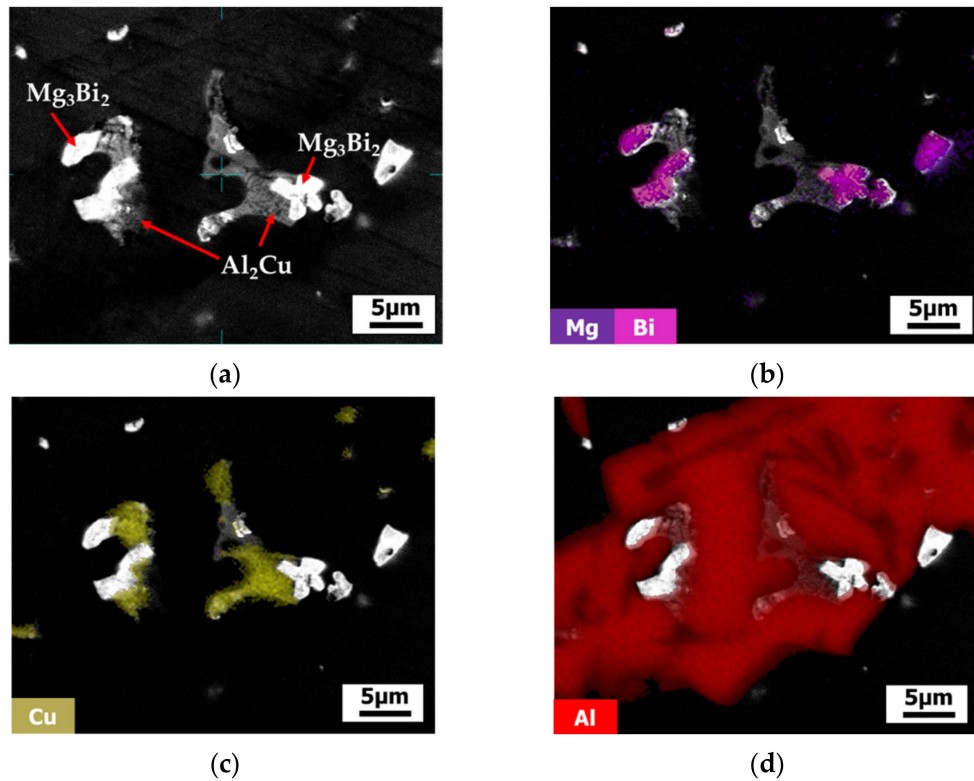

**Figure 12.** Elemental mapping shows on B390-1.0%wt Bi (**a**) Mg$_3$Bi$_2$ and Al$_2$Cu (**b**) elemental mapping of Mg and Bi (**c**) elemental mapping of Cu, and (**d**) elemental mapping of Al.

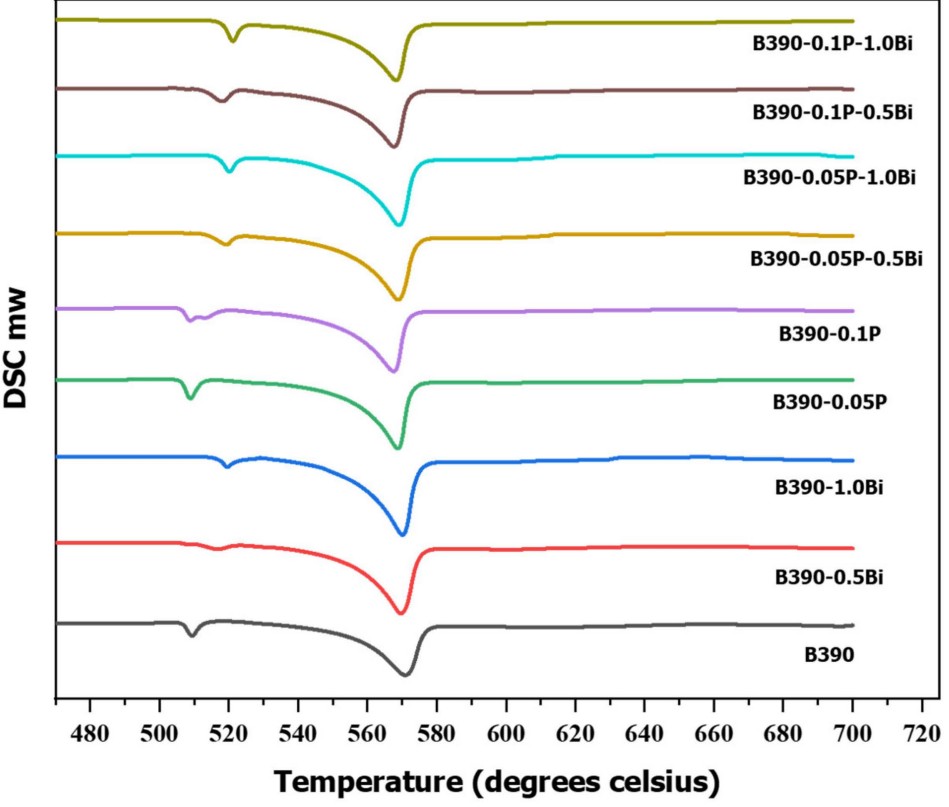

**Figure 13.** DSC curves at unmodified and modified as-cast alloys.

**Table 4.** Solidification temperature of the unmodified and modified alloys.

| Sample | Solidus Temperature (°C) | Liquidus Temperature (°C) | Solidification Range (°C) |
|---|---|---|---|
| B390 | 507 | 646.9 | 139.9 |
| B390-0.5Bi | 515.7 | 642.9 | 127.2 |
| B390-1.0Bi | 517.7 | 643.9 | 126.2 |
| B390-0.05P | 507.4 | 652 | 144.6 |
| B390-0.1P | 506.4 | 665.1 | 158.7 |
| B390-0.05P-0.5Bi | 513.5 | 650.7 | 137.2 |
| B390-0.1P-0.5Bi | 513.1 | 665.8 | 152.7 |
| B390-0.05P-1.0Bi | 517.3 | 650.9 | 133.6 |
| B390-0.1P-1.0Bi | 518.2 | 666.2 | 148 |

### 3.4. Effect of Bi and P on Hardness and Surface Roughness

### 3.4.1. Hardness Test

Figure 14 shows the hardness of the unmodified and modified alloys in their as-cast condition. It was observed that the hardness of unmodified alloys was 77.4 HRB, and then it increased to 78.23 HRB with the addition of 0.5 wt%Bi. This is due to the presence of a hard $Mg_3Bi_2$ phase. However, the hardness value slightly decreased as the concentration of Bi was increased to 0.1 wt% because the clustering of the primary Si structure resulted in a larger spacing between Si clusters. In contrast, the specimen with 0.1 wt%P, exhibited the highest hardness, 81.93 HRB as a result of a fine and uniform distribution of primary Si. This experimental result is consistent with the work of Maeng et al. (2000) [37] and Dang et al. (2017) [9]. Alternatively, when both P and Bi were added, the hardness value increased more than for the unmodified and Bi modified alloys, clearly demonstrating that P has an effect in enhancing hardness, whilst Bi has a negative effect.

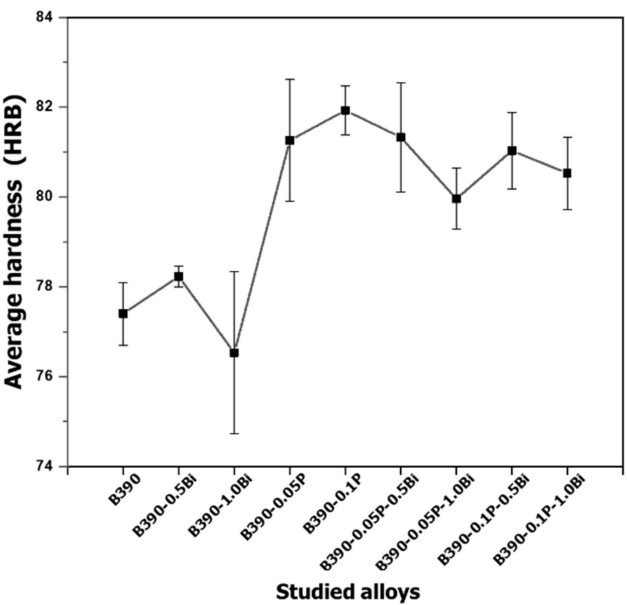

**Figure 14.** Average hardness of unmodified and modified as-cast alloys.

### 3.4.2. Surface Roughness

Figure 15 presents the surface roughness (Ra) values after cutting at various speeds for all alloys examined in the current study. The average surface roughness (Ra) of the unmodified alloy after being cut at speeds of 78 and 98 m/min was 5.51 μm and 5.65 μm, respectively. Increasing the cutting speed from 78 to 98 m/min had little effect on surface roughness because the unmodified alloys exhibited a large amount of primary Si and had a fibrous morphology in the eutectic silicon phase. For Bi modified alloys, increases in

the Bi content decreased surface roughness at a cutting speed of 78 m/min and effectively decreased the surface roughness by 10.16% and 18.15%, respectively, for 0.5 wt%Bi and 1.0 wt%Bi samples, compared to the unmodified alloy. Bi content produced a significant reduction, 58.8%, in surface roughness at high cutting speeds after 1.0 wt%Bi addition. This result is in agreement with the work of Barzani et al. (2013) [17]. There is a formation of fine lamellar structures along with the lubrication properties of Bi. For P addition, it was found that the surface roughness was slightly reduced compared to the unmodified alloy. However, when the cutting speed was increased to 98 m/min, the surface roughness was reduced by 42.83% and 48.50%, respectively, for 0.05 wt%P and 0.1 wt%P samples. This result indicates that P content has less effect on surface roughness of an alloy than its Bi content. Since the unmodified and P modified alloys exhibited eutectic fibrous silicon morphology that induced an edge build up, a negative effect on surface roughness values resulted [17].

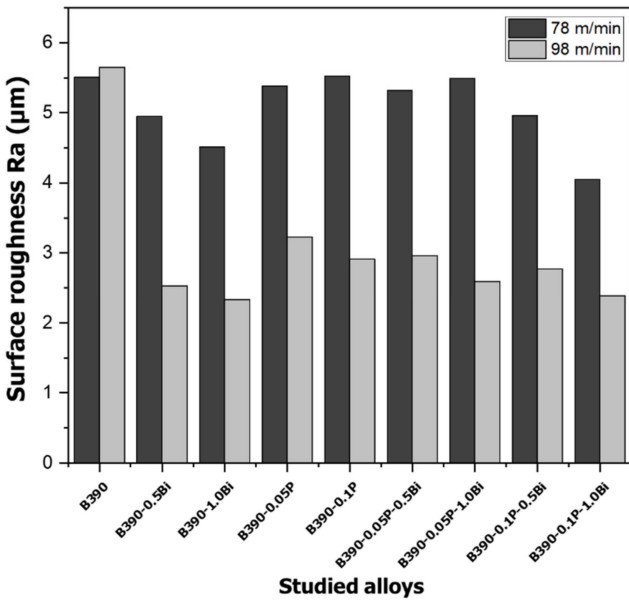

**Figure 15.** Average surface roughness Ra (µm) at unmodified and modified as-cast alloys.

In the cases of Bi-P modifications at a cutting speed of 78 m/min, surface roughness was slightly reduced, by up to 26.5% when the Bi content was increased to 1.0 wt%. At a cutting speed of 98 m/min, a significant improvement on surface roughness was also observed. Surface roughness decreased with increased Bi content. From the experiment, the best improvement was obtained from the B390-P0.1-Bi1.0 alloy. The experimental results show that the unmodified alloys exhibited a higher surface roughness than the modified alloys for all cutting conditions due to the size and distribution of primary Si and the fibrous morphology of the eutectic silicon phase. The modified alloys had smoother surfaces after being cut at 98 m/min. Therefore, surface roughness is clearly related to changes in the primary Si and eutectic Si phase morphologies. This facilitates cutting at higher speeds in practice, thereby increasing machining efficiency.

In this work, effect of $Mg_3Bi_2$ on machinability is not yet clearsince the amount of $Mg_3Bi_2$ phase is limited by the amount of Mg. An increase in Bi content led to an increase in the number of Bi particles that benefit machinability. $Mg_3Bi_2$ does not support machinability since $Mg_3Bi_2$ is hard and has a high melting point 823 °C [34,38].

The results of this experiment show that P and Bi addition benefits the aluminium B390 hypereutectic alloy by modifying both primary Si and the eutectic phase, especially in permanent mould or high pressure die casting applications. Although microstructural observation showed the best modification in B390-0.1P-0.5Bi for surface quality after high-

speed cutting of the B390 alloy, Bi can be added by up to 1.0 wt%. This amount of Bi showed a superior effect compared to Sr and Sb, for a hypoeutectic aluminium alloy [17].

## 4. Conclusions

From the experimental results, the following conclusions can be drawn:

(1)    P addition has a positive effect on the refinement of primary Si, while Bi addition has a negative effect. However, addition of both P and Bi can be used to reduce the size and distribution of the primary silicon phase compared to unmodified and Bi only modified specimens.

(2)    The morphology of eutectic Si can be changed from flakes and coarse fibres to an evenly distributed fine lamellar structure by the addition of both P and Bi. The P content induces a high concentration of silicon atoms around the primary Si. Additionally, a modified condition using P and Bi together reduces the sizes of intermetallic phase regions.

(3)    The Bi content shows no significant effect on hardness. However, P addition yields a significant improvement in the hardness value, 81.93HRB or a 5.85% increase. Adding Bi and P together to the melt at levels of up to 1.0 wt% does not degrade the effect of P on hardness improvement of the modified alloys.

(4)    P addition increases the liquidus temperature, leading to the wider solidification range. This is due to AlP particles acting as nuclei for primary Si at higher reaction temperatures. Alternatively, the reduction of solidus temperature owing to an increase in Bi content results in a narrower solidification range leading to a modified eutectic Si and intermetallic compounds. In the case where P and Bi were added to the melt together, both the solidus and the liquidus temperatures were increased, leading to modification of both the primary Si and eutectic Si phases.

(5)    Among the P-Bi modified alloys, the most effective reduction in roughness from cutting at both low and high speeds was for the B390 alloy modified by addition of 0.1 wt%P and 1.0 wt%Bi by 26.5% and 57.7%, respectively. This is largely due to morphology changes of eutectic silicon from a large bulky primary Si and flakes to a fine primary Si with a lamellar eutectic. Additionally, Bi also eases machining efforts by acting as a lubricant during cutting.

**Author Contributions:** Conceptualization, S.T. and P.J.; writing—original draft preparation, P.J.; investigation P.J.; writing—review and editing, S.T.; methodology, S.T. All authors have read and agreed to the published version of the manuscript.

**Funding:** The financial support was provided by the National Research Council of Thailand: NRCT (No. 6200087) and Khon Kaen University under the Graduate School: GS KKU.

**Institutional Review Board Statement:** Not applicable.

**Informed Consent Statement:** Not applicable.

**Data Availability Statement:** Not applicable.

**Acknowledgments:** This research was supported by Supply Chain and Logistics System Research Unit, and Research and Graduate Studies, Khon Kaen University.

**Conflicts of Interest:** The authors declare no conflict of interest. The funders had no role in the design of the study; in the collection, analyses, or interpretation of data; in the writing of the manuscript, or in the decision to publish the results.

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
