# Peer review of "Microstructural Modification Hardness and Surface Roughness of Hypereutectic Al-Si Alloys by a Combination of Bismuth and Phosphorus"

_crystals, doi:10.3390/cryst12081026_

Round 1

Reviewer 1 Report

This manuscript present original results of experimental research of the effects of bismuth and phosphorus on the microstructure, hardness and roughness of a B390 hypereutectic Al-Si alloy.

The most important are the following new results:

- Addition of the phosphorus less than 0.1 wt% shows a significant increase in the hardness value of B390 hypereutectic Al-Si alloy.

 - Addition of bismuth together with phosphorus (up to 1.0 wt %) did not reduce the effect of phosphorus on hardness of the modified alloys.

- The most effective reduction in roughness during cutting at both low and high speeds was received by modifying the B390 alloy using 0.1 wt of phosphorus with 1.0 wt % bismuth.

- The solidus temperature of the hypereutectic Al-Si alloy (B390 increased with bismuth content, resulting in a shortened solidification range.

- The primary silicon and eutectic silicon in B390 hypereutectic Al-Si alloy were modified when phosphorus and bismuth were added together to the alloys. Furthermore, an intermetallic phase size reduction also was received.

- The conclusions made in the manuscript are based on the analysis of the results of experimental studies of B390 hypereutectic Al-Si alloy ingots with phosphorus and bismuth additives. These ingots were obtained by the authors by the method of melting in an alumina crucible in a furnace

The research of this manuscript is in scope of the journal.

Research is good scientifically valid.

Methods and results are technically correct.

Results can be fully reproducible.

Author Response

Dear reviewer,

Thank you so much for your kindness and valuable comments and suggestions. 

Reviewer 2 Report

The presented manuscript is devoted to the study of the effect of co-modification with phosphorus and bismuth on the change in the size and morphology of the structural components of a B390 hypereutectic Al-Si alloy. The authors convincingly show the lack of knowledge about the joint modifying effect of phosphorus and bismuth on the structure of hypereutectic Al-Si alloys in the published literature. A good comparative analysis of existing publications and the tasks set in the work is given, and the topicality of the work performed is presented and described clearly. The methodological part of the manuscript is presented in sufficient detail. Large statistical samples confirm the reliability of the presented experimental data. In general, I believe this manuscript merits publication, but have a few comments and suggestions for the authors before publication:

1. The title of section 3.4 (as well as the title of the manuscript itself) does not fully correspond to its content, since only hardness is considered among the mechanical properties. At the same time, readers would like to see how co-modification affects such significant mechanical properties as tensile strength, yield strength, and ductility. 

2. I recommend adding some specific quantitative data to the abstract, as long as it contains only descriptive statements. Conclusions also should be specific, not declarative.

3. Before conclusions, it is recommended to briefly describe what practical application the obtained results have and how they can be used in real industrial production. 

4. Some minor mistakes have been detected in the text, it is recommended to revise and correct some grammatical errors. 

This work is definitely of interest to a wide audience in several fields and of an importance and novelty that warrants publication in Crystals. I recommend accepting this manuscript for publication after correcting some shortcomings. 

Author Response

Dear reviewer,

Thank you so much for your kindness and valuable comments and suggestions. All suggestions has been responsed in the attachment. 

Reviewer 3 Report

The authors have studied the the effects of bismuth and phosphorus on the microstructure and roughness of a B390 hypereutectic Al-Si alloy. Changes in the microstructure of the as-received alloy and its modified variants as well as morphology of microstructure components were estimated. Mechanical properties and machinability of the alloys were also evaluated. The authors do find some interesting results. The work is well done, but some deficiencies need to be corrected to make the manuscript acceptable for publication.

(1) A general comment: The authors should capitalize the first letter of the first word in each axis name in Figures.

(2) A general horizontal axis name like “Studied alloys” should be added below the row of specified alloy compositions for Figs. 4, 14, and 15.

(3) The authors should increase the font size for scale bars and other text in Figs. 2, 3, 5, 6, 7, 8, 9, 10, 11, 12, 13, 14, and 15.

(4) Fig.9(b),(d): A font size for “AlP” is too small to be recognized. Please, use external arrows and text to mark the particles.

(5) The authors should mention AlP particles while explaining primary Si particle formation from the melt (Lines 315–325), using Fig.9(b),(d). The authors may somehow combine this paragraph with one given earlier (Lines 240–247).

(6) Fig.13: The horizontal axis name should be as follows: “Temperature (degrees Celsius)” or “Temperature (°C)”.

(7) Fig.15: The vertical axis name should be as follows: “Surface roughness Ra (µm)”. The following explanation should be added to the figure caption: “Legend denotes corresponding cutting speed”.

(8) Section “Conclusions” (2), Lines 452–453, the following sentence should be corrected: “The P content continues to influence the concentration of silicon in the matrix”. Please replace the word “influence” with a more certain word or clarify this influence.

Author Response

Dear reviewer,

Thank you so much for your kindness and valuable comments and suggestions. Authors response is in the attachment.

Round 2

Reviewer 3 Report

(1) In my opinion, linguistic errors present in the title of the manuscript should be corrected. Maybe, the authors can add a few words as follows: “Microstructural Modification and Improvement of Hardness and Surface Roughness of Hypereutectic Al-Si Alloys by a Combination of Bismuth and Phosphorus”

(2) Line 21: The authors missed a dot between sentences.

(3) Line 316: The authors should remove a bracket after the phrase “for primary Si”.

All comments of the reviewer were taken into account. The manuscript can now be published in Crystals.